# Data Augmentation Techniques for Accurate Action Classification in Stroke Patients with Hemiparesis

**DOI:** 10.3390/s24051618

**Published:** 2024-03-01

**Authors:** Youngmin Oh

**Affiliations:** School of Computing, Gachon University, Seongnam 13120, Republic of Korea; youngminoh@gachon.ac.kr; Tel.: +82-31-750-5795

**Keywords:** data augmentation, action recognition, wearable sensors, stroke rehabilitation

## Abstract

Stroke survivors with hemiparesis require extensive home-based rehabilitation. Deep learning-based classifiers can detect actions and provide feedback based on patient data; however, this is difficult owing to data sparsity and heterogeneity. In this study, we investigate data augmentation and model training strategies to address this problem. Three transformations are tested with varying data volumes to analyze the changes in the classification performance of individual data. Moreover, the impact of transfer learning relative to a pre-trained one-dimensional convolutional neural network (Conv1D) and training with an advanced InceptionTime model are estimated with data augmentation. In Conv1D, the joint training data of non-disabled (ND) participants and double rotationally augmented data of stroke patients is observed to outperform the baseline in terms of F1-score (60.9% vs. 47.3%). Transfer learning pre-trained with ND data exhibits 60.3% accuracy, whereas joint training with InceptionTime exhibits 67.2% accuracy under the same conditions. Our results indicate that rotational augmentation is more effective for individual data with initially lower performance and subset data with smaller numbers of participants than other techniques, suggesting that joint training on rotationally augmented ND and stroke data enhances classification performance, particularly in cases with sparse data and lower initial performance.

## 1. Introduction

Stroke survivors often experience hemiparesis, which is characterized by weakness or mobility limitations on one side of the body [1]. Many post-stroke patients experience hemiparesis owing to incomplete recovery of motor function in the upper extremities [2]. To restore motor function in the affected limb, patients require extensive training involving repetitive movements using the use-dependent learning mechanism to facilitate neural plasticity [3]. However, upon discharge from the clinic, most patients tend to develop a habit of not using the affected limb in their daily lives, motivated by the fear of negative consequences such as pain and the failure of motor tasks—this phenomenon is known as learned non-use [4]. Encouraging the use of the affected limb in daily activities requires a feedback system that monitors the patterns and frequencies of use [5]. Wearable sensors, such as smartwatches and smart bands, are suitable hardware candidates for feedback systems because of their widespread availability. For home-based rehabilitation systems to perform effectively, they must utilize classification algorithms capable of recognizing the activities of daily living (ADL) performed using the affected side.

With recent advances in deep learning, several operational models for recognizing activities within non-disabled (ND) populations have been proposed [6,7]. Deep-learning models do not depend on domain-specific feature engineering; instead, they automatically extract learned features, rendering them generalizable to other domains. However, the distinct patterns and sparsity inherent in movement data of stroke patients complicate the direct application of deep learning methods to activity recognition.

While wearable sensor data have been primarily utilized for the diagnosis and assessment of patient movement [8,9], several studies have explored methods to enhance the performance of classification models by incorporating data from both ND populations and patients with movement disabilities, e.g., hemiparetic stroke patients. O’Brien et al. [10] used smartphone sensors to train a random forest (RF) model to classify four gait-related activities (lying and sitting, standing, walking, and using the stairs) in both gait-impaired patients with stroke and healthy participants. The study revealed that a classifier trained on data gathered from healthy subjects performed poorly when tested on stroke data. Lee et al. [5] utilized bilateral inertial measurement unit (IMU) sensors on both wrists to classify ADL tasks in stroke survivors and a control group. ADL movements were classified as either goal-directed or non-goal-directed using logistic regression and the kinematic characteristics of the movements were estimated using a RF model. Meng et al. [11] utilized accelerometers, gyroscopes, and surface electromyography (EMG) signals to classify four ADL movements (walking, brushing teeth, washing face, and drinking) in stroke patients and ND individuals. Various machine learning-based models were trained, with a support vector machine exhibiting the best performance. The study by Um et al. [12] is one of the few studies to have applied various time-series transformations, such as scaling, rotation, permutation, magnitude, and time-warping, to augment wearable sensor data; the augmented data were used for training a binary classifier to distinguish bradykinesia from dyskinesia in Parkinson’s disease. It demonstrated that most transformations enhance the classification performance of a convolutional neural network (CNN) model, with rotation, permutation, and time-warping proving to be the most effective. Celik et al. [13] converted IMU data into images and augmented to improve the classification performance of pre-trained CNN models. The data consisted of six classes (walking, ascending, descending, sitting, standing, and lying down) sourced from healthy subjects, individuals with Parkinson’s, and stroke survivors.

However, most existing studies have focused on data related to only a few gross body movements, typically involving walking, standing, and lying. To assess the impact of data augmentation on real-world applications, we conducted experiments encompassing 52 ADL movements involving the upper extremities in a real home environment [14]. In this prior study, we discovered that joint training using data from both ND individuals and stroke patients yielded more effective classification performance compared to training based on either ND or stroke data exclusively. Additionally, the study demonstrated that data augmentation based on rotation enhanced the model performance significantly under all tested conditions. However, numerous research questions remain unanswered, such as the amount of augmented data required, the effectiveness of other types of transformations, and the influence of different augmentation strategies on the evaluation of individual data.

In this context, the current study aims to investigate the effects of data augmentation systematically with the following goals. First, data augmentation based on rotation, permutation, and time-warping was tested using two different baseline datasets involving varying amounts of synthetic data—stroke data and ND + Stroke data. Second, transfer learning based on the pre-trained ND data and training using a more advanced convolutional model, InceptionTime [15], were tested with rotation augmentation. Third, the classification performance of individuals in the stroke group while using rotation augmentation was investigated. Finally, subsets with different numbers of participants, ranging from two to 12, were sampled from the original stroke data, and the effect of rotation augmentation was measured to evaluate the contribution of data augmentation to the sparse data domain.

## 2. Materials and Methods

This study investigates the effects of training strategies data augmentation techniques on activity recognition in stroke patients, where training data are sparse and exhibit a distinct distribution compared to ND populations. To achieve this goal, various data transformations and training strategies, including transfer learning and joint training, were evaluated on ADL tasks based on the Jeonju University Inertial Measurement Unit (JU-IMU) dataset. This dataset is publicly available from a previous study [14] “https://github.com/youngminoh7/JU-IMU (accessed on 28 February 2024)”.

### 2.1. Dataset

The publicly available JU-IMU dataset introduced in a previous study was used. Although the dataset also included the range of motion task, only the ADL task was used for training and evaluation in this study. In aggregate, 42 participants took part in the study, including 28 ND volunteers and 14 patients with hemiparetic stroke. The participants performed 52 different ADL movements in a home environment. The dataset comprised values of five IMU sensors attached to both sides of the wrists, upper arms, and trunk of the participants. Nine out of the 14 stroke subjects suffered from right hemiparesis (subgrouped as “StrokeRH”), while the remaining six suffered from left hemiparesis (“StrokeLH”). The patients were instructed to perform ADL movements primarily on their affected side, even though all of them were right-handed before the onset of stroke. Severity was measured using various clinical assessments, with the Fugl-Meyer Assessment for Upper Extremity (FMA-UE) being a representative clinical metric. The mean and standard deviation were 50.4 and 10.2, respectively, with the minimum and maximum values among the patients being 33 and 66, respectively, out of a total of 66 points. Detailed information on the dataset is provided in the Materials and Methods section of a previous study [14].

### 2.2. Preprocessing

In the ADL task, 42 participants performed each of the 52 movements five times consecutively. The segments of the five repeated movements were annotated by human experimenters. A single segment of data consisted of 30 channels corresponding to five sensors equipped with six channels each (*x*, *y*, and *z* axes of an accelerometer and gyroscope). Before the segments were transmitted into deep learning models, their lengths were adjusted to be exactly 3700 time points using linear interpolation, corresponding to a 46.25-s duration. Additionally, all channel-wise means were subtracted from the raw signals to make the mean of each channel in the segments equal to zero. Scaling was applied to match only the between-sensor scales because the accelerometer and gyroscope signals exhibited different units and scales. To this end, all accelerometer signals were divided by 2.537, and all gyroscope signals were divided by 0.478. These divisor constants were derived based on the global standard deviation of all corresponding channels across the segments. Finally, all channel signals were smoothed by taking the moving average of the last ten points. These preprocessed segments were then sliced into multiple windows by applying the sliding window technique with a window size of 740 and stride of 150 time points. Twenty windows were produced for each segment, serving as unit data for the classification models.

### 2.3. Models

The same Conv1D baseline model in the previous study [14] (Figure 1a) was adopted. It comprised four convolutional layers, with 32, 64, 128, and 256 kernels with one-dimensional size = 5 and stride = 2. The kernels were convolved using the temporal dimensions of the data. After passing through the convolutional layers, the output was flattened and transmitted into three dense layers with 800, 200, and 52 neurons, respectively. The rectified linear unit (ReLU) activation function [16] was applied following each convolutional or dense layer. Additionally, a dropout operation [17] with a probability of 0.7 was applied to the first two dense layers to regularize overfitting.

In addition to the baseline Conv1D model, the more advanced InceptionTime model [15], based on the Inception module (Figure 1b), was trained and evaluated in some of the augmentation scenarios (Section 2.5). The Inception module reduces the number of channels by applying a bottleneck layer following a convolution with a kernel of size 1. From the bottleneck layer, three kernels of different sizes (10, 20, and 40) were convolved through the temporal dimension. Additionally, the maximum pooling of the input was convolved with a kernel of size 1 and concatenated with the output of the three kernel convolutions. This inception module captured temporal patterns of various lengths effectively. The output of the Inception module was transmitted to the subsequent Inception module stacked with the residual connections. There were six Inception modules in aggregate, each equipped with 32 filters. In both models, the final layer was a softmax layer with 52 neurons, corresponding to the number of classes in the ADL task.

### 2.4. Data Augmentation

Among simple time-series transformations techniques, three methods known to be effective in boosting classification performance [12]—rotation, permutation, and time-warping—were tested in this study. The rotation transformation rotated a vector formed by the three axes of either the accelerometer or the gyroscope in three-dimensional space. Components of a rotation axis vector were randomly sampled from a uniform distribution over [−1, +1], and the rotation angle was randomly sampled from a uniform distribution over [−90°, +90°]. Rotation was applied to all time points in the data window. Permutation transformation was used to split each data window into four subsegments of different lengths and randomly permutate the temporal orders to create a new window. Time-warping transformation was used to distort the temporal intervals between time points randomly. Figure 2 depicts examples of these three transformations applied to a data sample. Random seeds were used to control the randomness in the transformations.

### 2.5. Training and Evaluation

#### 2.5.1. Leave-One-Subject-Out Cross-Validation (LOSOCV)

To ensure unbiased estimation of the effects of data augmentation, leave-one-subject-out cross-validation (LOSOCV) was employed under all conditions. Two conditions using only un-augmented original data, the stroke condition (Figure 3a, without using transformed data) and ND + Stroke condition (Figure 3b, without using transformed data) were used as baselines. For each baseline condition, a set of transformed data using one of the three transformation methods (rotation, permutation, and time-warping) was added to the original training data for augmentation. To observe whether more augmented data would improve model performance, both single and double augmentation strategies were applied. Single augmentation involved the same amount of data as the original stroke training data (from 14 − 1 = 13 participants in each cross-validation). Double augmentation involved two sets of transformed data obtained from the stroke training data. These two sets of transformed data differed from each other in terms of the sets of seeds applied to control the randomness. Additionally, to evaluate whether the transformed data preserve the properties and features of the original data, the Train on Synthetic, Test on Real (TSTR) evaluation method [18] was applied. In this approach, LOSOCV was trained solely using the transformed Stroke data, with the original Stroke data reserved for evaluation.

Data augmentation was based solely on the Stroke data, leaving the ND data unchanged. This was intended to maintain a balance between the two distinct datasets of ND and Stroke because the amount of available stroke data was only half of that of available ND data. The effect of data augmentation is significantly influenced by the specific instance of the transformed data used; hence, multiple LOSOCV runs (three to nine) were applied for each combination of the original data (Stroke vs. ND + Stroke) and their transformed forms (rotation, permutation, and time-warping). In each LOSOCV run with data augmentation, different sets of seeds were used to generate different instances of transformed data.

#### 2.5.2. Training with the Stroke Subsets

To examine the impact of data augmentation in sparse data scenarios, where only a limited number of Stroke patient data are accessible, we subsampled various numbers of participants from the Stroke group. Out of 14 Stroke patients, *n* participants (where *n* = 12, 10, 8, 6, 4, 2) were randomly selected and the data of these participants constituted training subsets. The subsets of *n* = 2, 3, and 6 were each sampled six, four, and two times, respectively, to mitigate biases stemming from small sample sizes. Evaluation was conducted on all Stroke participants not included in the training subsets. The evaluation results were averaged across all test participants, similar to LOSOCV. The single and double rotation transformations were applied to all the training subsets.

#### 2.5.3. InceptionTime and Transfer Learning

The aforementioned LOSOCV runs were tested using the baseline Conv1D model to ensure consistency. Additionally, the InceptionTime model introduced in Section 2.3 was also tested with the single and double rotational augmentations under the ND + Stroke condition. Furthermore, transfer learning was carried out by re-instantiating the final dense layer before the softmax layer in the Conv1D model. In this transfer learning scenario, the model was initially trained solely on the ND data. Subsequently, the final dense layer was re-instantiated with random parameters, and the parameters of the remaining layers were set to be updated based on additional training using the Stroke data, both with the original and the rotation-augmented data.

#### 2.5.4. Optimization

For optimization, the AdamW optimizer (learning rate = 0.001, betas = (0.9, 0.999), eps = 1 × 10^−8^, and weight decay = 0.01) [19] was utilized to minimize the cross-entropy loss. The batch size was set to 256, and the training epoch was adjusted to comprise approximately 5200 batch iterations. Because the augmented conditions corresponded to more training data than the original, the adjusted training epoch ensured that all LOSOCV conditions had similar number of iterations during training. The final result was evaluated as the median F1-score between fixed iterations ranging from 5000 to 5009.

### 2.6. UMAP

To evaluate the contribution of data augmentation, particularly rotation transformation, to the enhancement of classification performance, a dimension-reduction technique called uniform manifold approximation and projection (UMAP) [20] was applied. UMAP is based on Riemannian geometry and algebraic topology. Similar to t-SNE [21], it projects high-dimensional data distributions to lower-dimensional spaces; however, is more scalable to larger datasets, offering increased run-time performance. The technique was applied to the output feature of the final dense layer before the softmax layer in the Conv1D model, mapping from a domain of dimension 200 to one of dimension 2 for visual inspection. In this manner, the model’s learned mappings of data distributions can be visualized by projecting all data instances. The projected distributions of the original stroke and rotation-augmented data were compared visually.

### 2.7. The OPPORTUNITY Dataset

The OPPORTUNITY dataset [22] contains sensor recordings of daily activities in a realistic environment similar to the JU-IMU dataset. To evaluate the effectiveness of data augmentation methods on this publicly available dataset, we adopted the OPPORTUNITY challenge setup, where four participants engaged in 18 kitchen activities. We selectively chose IMU sensors attached to the upper body and both feet, resulting in a dataset with 54 dimensions. The NULL class in the original dataset was excluded to apply data augmentation. Besides the sensor dimension and the exclusion of the NULL class, the remaining training and test protocols followed those of the DanHAR model [23], one of the state-of-the-art (SOTA) methods for the OPPORTUNITY challenge. Additionally, the InceptionTime model was also trained and tested for reference. The three data augmentation techniques—rotation, permutation, and time warping—were applied to the predefined split of training data. Unlike JU-IMU, however, both the training and test data contained split subsets from the same subjects (S2 and S3). The OPPORTUNITY dataset contained imbalanced class distributions, prompting the use of data augmentation to bolster the classes with fewer samples. As a result of data augmentation, all classes except one were augmented to have 2500 training samples. The exceptional class had 2583 samples, with no additional augmented data applied to it.

## 3. Results

### 3.1. Data Exploration

Figure 4 depicts an example of individual variability within the same class (“InsertEnvelope”) and Figure 5 shows overlapping plots of the same class. The top two rows depict data from eight participants in the ND group and the bottom two rows depict data from eight participants in the stroke group. Notably, the two patterns corresponding to the ND and Stroke groups were distinct. Participants in the stroke group exhibited less prominent peaks in terms of both amplitude and duration than those in the ND group. The mean curves in Figure 4 exhibit similar patterns. Additionally, the individual signals were much less coherent in the stroke group than in the ND group (mean correlation coefficient between signals: stroke = 0.048, ND = 0.226), resulting in an even weaker mean signal than that of the ND group.

### 3.2. Training Results

#### 3.2.1. Baseline Results

Figure 6 depicts the learning curves of all participants with stroke in the two baseline LOSOCVs—one trained with only stroke data and another trained using ND + Stroke data. Each curve depicts an increase in the participant’s F1-score when the baseline Conv1D model was trained using the other participants’ data. Table 1 summarizes the training results. The mean F1-score for each condition was calculated by averaging the participants’ individual F1-scores. Each participant’s individual F1-score was obtained from the median of the F1-scores between 5000 and 5009 iterations. The F1-scores of the Stroke group increased when trained with the ND + Stroke data (54.0%) compared to the stroke-only data (47.3%). However, StrokeRH (56.9%) increased much more than StrokeLH (48.7%). Note that the participants in both the ND and StrokeRH groups used their right hand, whereas those in the StrokeLH group used their left hand as the primary hand while performing asymmetric activities.

#### 3.2.2. Data Augmentation

Three types of data transformations—rotation, permutation, and time-warping—were tested for both Stroke and ND + Stroke baselines. Figure 7 and Table 2 compare the effects of the data augmentation techniques in single- and double-augmentation configurations. Nine LOSOCV runs were performed for each augmentation configuration with different random seeds. The colored semi-transparent dots in Figure 7 represent the cross-subject mean F1-scores corresponding to each LOSOCV run, calculated in the same manner as the baselines. In aggregate, nine (some overlapping) colored dots were obtained for each augmentation method. The lines in Figure 7 connect the baseline to the mean F1-scores corresponding to each augmentation method. Under both baseline conditions, data transformations were applied only to the Stroke data. Among the three augmentation transformations, rotation yielded the greatest increase in F1-score, followed by time-warping and permutation, both in the Stroke and ND + Stroke baselines. Double augmentation yielded only a slight increase in performance compared to single augmentation. Permutation yielded only a small increase with single augmentation and a slight decrease with double augmentation. Overall, the double rotational augmentation on the ND + Stroke baseline yielded the best F1-score of 60.9% among all augmentation conditions when trained with the Conv1D model.

Table 3 displays the outcomes of the Train on Synthetic, Test on Real (TSTR) [18] evaluation method. The three transformed datasets (rotation, permutation, and time warping) comprised exclusively of transformed signals as training data, while the original data was reserved solely for evaluation. The transformed training data all demonstrated slight increase in F1-scores compared to the original data, suggesting that these transformations preserve the characteristics of the original data and may even enhance generalizability to unseen data.

Figure 8 and Table 4 present the results for two additional training methods—transfer learning using the pre-trained model (Conv1D) and training using the InceptionTime model. First, the pre-trained model was trained using all 28 ND participants’ data and then it was fine-tuned using the data of participants with stroke, except one left for validation, following the LOSOCV approach. The resulting F1-scores of transfer learning were slightly lower than, albeit close to, those of the ND + Stroke Conv1D model, in which both ND and stroke data were used for training from the beginning. The InceptionTime model exhibited the best performance among all augmentation conditions considered. These additional training conditions also yielded an increase in the F1-score with rotational augmentation.

#### 3.2.3. Correlations of Individual Data

Figure 9 depicts the correlation between the FMA and baseline F1-scores of all participants in the stroke group. When training was conducted based solely on stroke data, the correlation coefficient was 0.63 (*p*-value = 0.016), and when it was conducted using ND + Stroke data, it was 0.75 (*p*-value = 0.002). These strong correlations between the functional assessment of a patient’s movement capabilities and the performance metric of the classification model indicate that the movement patterns of patients with more severe symptoms were more distinct from those of ND populations, thus leading to lower performance metric scores.

Figure 10 depicts the effects of rotational augmentation on an individual basis by calculating the percentage increase in F1-score compared to the baselines. The negative correlation (−0.70, *p*-value = 0.005) in the stroke baseline indicates that rotational augmentation further increased the F1-scores of participants with lower initial performances. The correlation in the ND + Stroke baseline was also negative (−0.49, *p*-value = 0.08), but its trend was less salient and the statistical significance was marginal. Higher initial F1-scores of the ND + Stroke baseline could have affected this observation.

#### 3.2.4. Training on Subsets

Figure 11 depicts the change in F1-score with respect to varying numbers of participants in the stroke training group. Corresponding to each number of participants, multiple subsets were sampled and trained, as in the case of LOSOCV. All stroke participants, except those included in the training subset, were evaluated for each condition. Multiple LOSOCV runs were averaged to observe the trends. Rotational augmentation was applied only to the training subsets. Corresponding to all participant numbers, the augmentations improved F1-scores by similar margins, with an average improvement in absolute value by 5.8% for single rotational augmentation and 6.9% for double rotational augmentation. However, the percentage increase relative to the original subsets was larger corresponding to subsets with fewer participants (right panel in Figure 11). This indicates that rotational augmentation was particularly effective when the amount of training data was small.

#### 3.2.5. Dimension Reduction

Figure 12 compares the UMAP embeddings of the original data (left panel) with those of the rotationally augmented data (right panel). Embedding was performed from the final dense layer before the softmax layer in the Conv1D model, projecting 200-dimension feature vectors onto a two-dimensional space. Each dot represents an instance of 15,640 stroke data points. Rotational augmentation shifted the distribution of the embedding and improved the separation of between-class clusters, thereby improving model performance.

#### 3.2.6. The OPPORTUNITY Dataset

Table 5 presents a summary of the mean F1-scores achieved through data augmentation on the OPPORTUNITY dataset using the two state-of-the-art (SOTA) models, DanHAR [23] and InceptionTime [15]. The results indicate that time warping effectively improves the F1-score, while permutation and rotation exhibit similar or slightly inferior performance. Given that training and evaluation were conducted on sliding windows generated from continuous and repetitive signals, the impact of permutation may already have been accounted for. Additionally, the inferior performance of rotation implies that the original data likely maintained consistent angular configurations over time. This observation may be attributed to the OPPORTUNITY challenge’s limited subject numbers, which comprised data from only five participants, with two participants (S2 and S3) contributing to both training and test sets. Consequently, even minor angular deviations from the original data could have compromised the model’s classification performance.

## 4. Discussion

In this study, the fundamental research questions were why the classification performance trained on the stroke participants’ data was much worse than that on the ND participants’ data, how they differed, and whether various training strategies, including data augmentation, transfer learning, and more advanced classification models, would improve performance.

Data exploration revealed that stroke data yielded less salient signals in time series in both spatial and temporal dimensions. This indicates that it is more difficult for a classifier to capture representative patterns from stroke data. Additionally, the stroke data exhibited large inter-subject variability in terms of performance, which rendered the identification of common features that work for all cases more difficult. Moreover, another variability in the stroke group arose from data asymmetry. Based on the experimental design, patients with left hemiparesis (StrokeLH subgroup) were instructed to use their affected, non-dominant hand (the left hand) primarily for asymmetric ADL movements. This instruction was implemented because, otherwise, the StrokeLH participants would use their non-affected (right) side, revealing no difference compared to the ND group corresponding to movements performed primarily using their non-affected side. Because of this, the StrokeLH participants experienced greater difficulty in moving their affected and non-dominant extremities, thereby creating distinct patterns compared to the StrokeRH data. Adding the ND group data improved the mean F1-score up to 54.0% from 47.3% in the case of the stroke-only baseline training despite their different patterns compared with that of the stroke group. However, the increase was mostly observed in the StrokeRH subgroup (56.9%); it was minimal in the StrokeLH subgroup (48.7%). This implies that the ND data overlapped more with the StrokeRH data, in which participants primarily used the same side.

Among the three data augmentation transformations tested in this paper, rotational transformation yielded the best results, followed by time-warping and permutation. This result was consistent across the two baselines (Stroke and ND + Stroke) as well as single and double augmentations (Figure 7 and Table 2). Relative to the performance enhancements for the baselines, the variability induced by different augmentation techniques was relatively small (less than 1% of the standard deviation compared to an approximately 6% increase in the absolute F1-score corresponding to rotation). However, the additional improvement achieved using double augmentation was minute compared to single augmentation corresponding to all transformations. This indicates that simple transformations have limitations in extending data distributions to further enhance classifier performance. The performance enhancement induced by rotation could be interpreted as a transformation simulating the variability in limb orientation of the participants during movements, supplementing the uncovered feature space from the original data. This was observed indirectly in the embedding space via dimension reduction using UMAP (Figure 12). However, the two transformations acting on the temporal dimension, permutation and time warping, were relatively less efficient. Permutation mimics partially mixing the order of movement segments, while time warping simulates lengthening or shortening the duration of sub-movements. These properties could already be inherent in the temporal patterns of the original data. Additionally, the sliding windows take the form of temporal transformation starting at different phases of repeated movements.

One notable observation was that the individual F1-scores of the participants exhibited a strong correlation with their functional assessments of movement capabilities (FMA). Thus, a higher F1-score in terms of model classification could indicate that a patient’s movement patterns are closer to those of the ND population and vice versa. The correlation coefficient for the ND + Stroke baseline (0.75) was greater than that for the Stroke baseline (0.63), which verifies this theory because joint training with the ND data further enhanced the data of the participants with higher FMA scores. Combined with rotational augmentation, the relative increase in F1-scores exhibited the opposite trend—participants with lower FMA scores exhibited further improvements compared to the baselines (Figure 10). This result, in conjunction with the test on subsets (Figure 11), suggests that the data of participants with lower initial performance usually exhibit a larger relative improvement compared to those with initially higher performance. Clinically, the correlation between F1-score and FMA may be utilized to assess a stroke patient’s motor functions in ADL at a home environment. While some studies have proposed methods for automatically evaluating a stroke patient’s motor functions from sensor data [9], few have investigated the connection between motor functions and the performance of a classification model in the domain of ADL. In addition to its original intent of contextualizing a home rehabilitation system, monitoring the individual classification performance may offer insights into the recovery of motor functions during home training.

Among the two new training strategies considered, the one based on transfer learning outperformed the stroke baseline but yielded slightly worse performance than the ND + Stroke baseline. The results indicate that transfer learning in this data configuration did not outperform joint training of the two populations. In contrast, training with InceptionTime yielded results superior to those obtained using other training strategies. InceptionTime includes a bottleneck layer, followed by kernels of three different lengths (10, 20, and 40), which enables the capture of temporal patterns at multiple scales over a long range using a relatively smaller number of trainable parameters. This efficiency of the model structure enabled the capture of features from sparse and varying data of the stroke group.

A few studies [12,13] have adopted an approach similar to that used in this study. Um et al. [12] applied various time-series transformations to augment movement data related to Parkinson’s disease to enhance a binary classifier’s performance in distinguishing bradykinesia and dyskinesia. Celik et al. [13] converted IMU data into images and subjected them to data augmentation to improve the classification performance of pre-trained CNN models. However, these studies tested data augmentation under restricted conditions, with conclusions that may not be generalizable to a broader context. Additionally, few studies have systematically investigated various combinations of training conditions to examine the effects of data augmentation. As a different approach to data augmentation, recent studies have adopted deep generative models to augment time-series data and improve classification performance [24,25]. However, these generative models require the generation of conditional synthetic data for augmentation, thereby requiring a substantial amount of labeled data for training. This limits the usability of generative model-based approaches in sparse data domains, such as the classification of patient movements.

## 5. Conclusions

The current study explored various training techniques, including data augmentation, to improve classification performance of stroke data. One hypothesis was that the abnormal movement patterns of stroke patients would result in distinct, albeit partially overlapping, feature distributions compared to ND populations performing the same movements. The effectiveness of the spatial transformation (rotation) in improving classifier performance suggests that augmenting data by covering unexplored feature spaces (see Figure 12) is an efficient strategy for addressing data heterogeneity. The study comprehensively investigated different training strategies and models, combination of original and transformed data volumes, and the impact of data augmentation at the individual subject level. These findings offer insights into enhancing movement classifiers to support development of home rehabilitation systems. The subset analysis further demonstrated that data augmentation also benefits sparser datasets, a crucial aspect of clinical applications where patients’ data collection is challenging.

However, the study has certain limitations. The methods were tested only on a single specific dataset (JU-IMU), partially due to the difficulty of accessing other public time series datasets on ADL movements with sufficient class numbers. Additionally, despite of improvement with various strategies, there remained a significant gap between the performance of classifiers on ND and stroke data. Further investigation into data augmentation techniques, including those derived from generative models [24,25,26] and utilizing datasets from other domains, is necessary to address the issue of data sparsity. Lastly, monitoring individual patient’s movement patterns in long-term rehabilitation may offer insights into the relationship between motor function recovery and changes in movement patterns, as well as the corresponding classification performance.

## Figures and Tables

**Figure 1 sensors-24-01618-f001:**
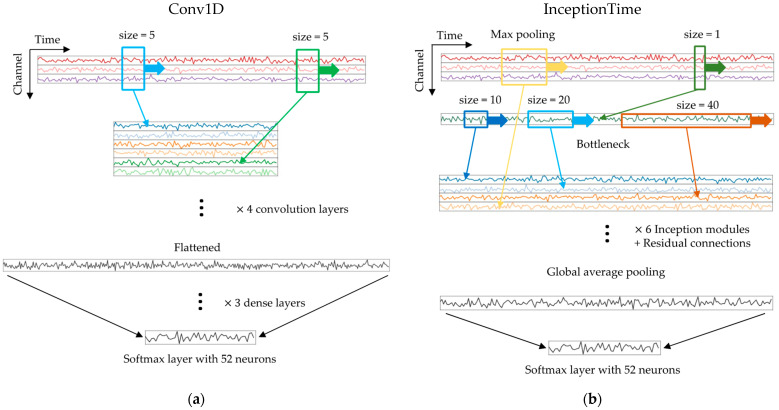
Schematic diagrams representing the data dimensions of the two deep learning models. The lengths of data and the number of channels do not match those in the real models used. Coloured lines indicate the result of a convolution operation. (**a**) Conv1D model: Kernels of fixed size perform convolution operations through the time dimension. Only two kernels are depicted for brevity. (**b**) InceptionTime model [15]: Number of channels is decreased via projection onto a number of “bottleneck” layers (only one bottleneck layer is depicted). From the bottleneck layers, one-dimensional kernels of different sizes are applied, and the resulting features are concatenated with the max pooling result.

**Figure 2 sensors-24-01618-f002:**
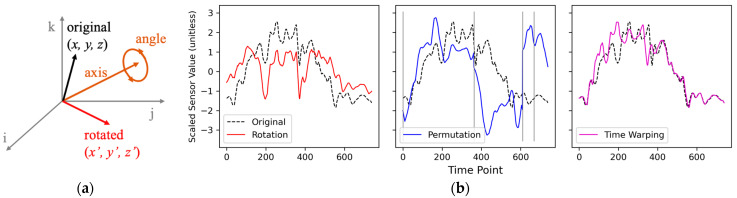
Data transformation for augmentation. (**a**) Demonstration of rotation at a certain time *t*. The three axes (*x*, *y*, and *z*) of a sensor form a vector (“original”) in a three-dimensional space. The vector is rotated by a random angle around a randomly chosen axis, resulting in a new vector (“rotated”) with the updated components (*x’*, *y’*, and *z’*). The rotation is consistently applied to all points overtime for a given data instance. Adapted from Ref. [14]. (**b**) Transformation examples. All plotted lines were obtained from the *x*-axis of the accelerometer in the sensor attached to the right wrist. (**Left** panel): rotation; (**Middle** panel): permutation; (**Right** panel): time-warping. Borders of the sub-segments are plotted as vertical gray lines.

**Figure 3 sensors-24-01618-f003:**
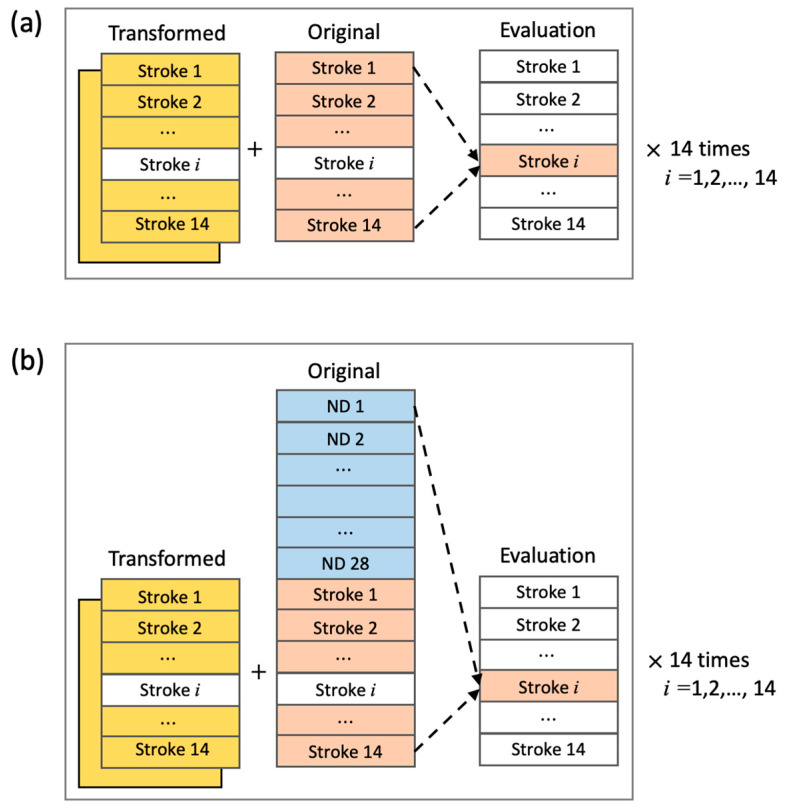
Training and evaluation based on LOSOCV. Participants’ data in Transformed and Original were used for training. Participants’ data with white background were not used for training. (**a**) The Stroke condition: Only Stroke data were used for both baseline training and augmentation training. In baseline training, no transformed data were used. (**b**) The ND + Stroke condition: All participants’ data in the ND group were used in conjunction with Stroke data. In baseline training, no transformed data were used. In augmentation training, transformed data from Stroke data were added together.

**Figure 4 sensors-24-01618-f004:**
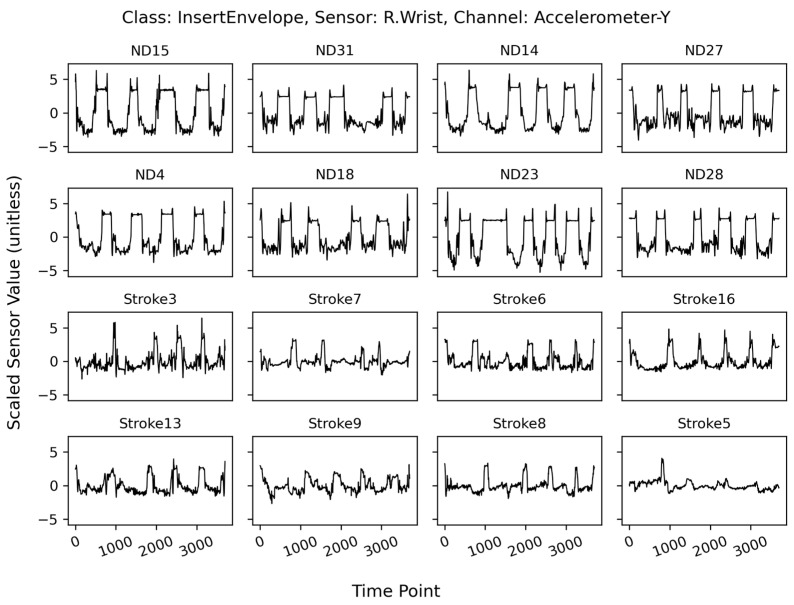
Example plots of a single channel (R.Wrist-Accelerometer-Y) corresponding to the class “InsertEnvelope” as obtained from 16 randomly selected participants—the first eight from the ND group and the last eight from the Stroke group. All sensor values were scaled in magnitude and interpolated to have the same duration (3700 time points). Each panel depicts five repetitions of the same movement.

**Figure 5 sensors-24-01618-f005:**
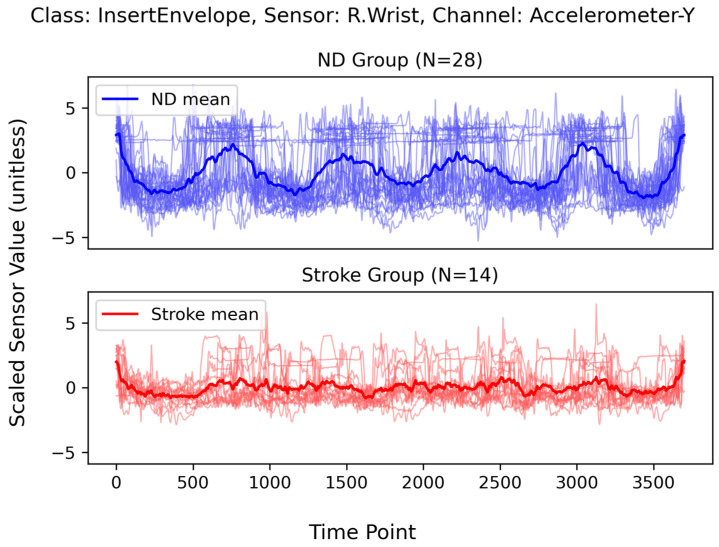
Overlapping plots of a single channel (R.Wrist-Accelerometer-Y) corresponding to the class “InsertEnvelope” as obtained from all participants in each group. The thin lines represent individual signals, and the thick lines represent the mean signal of each group. Top panel: The ND group. Bottom panel: The Stroke group.

**Figure 6 sensors-24-01618-f006:**
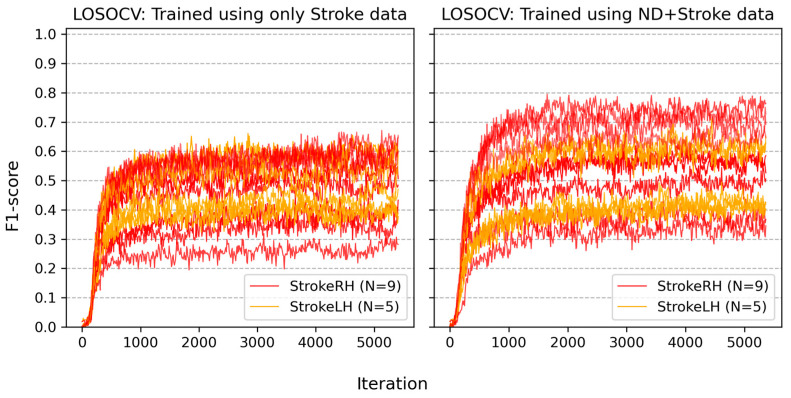
Learning curves of all stroke participants in two baseline LOSOCVs. Each curve represents the increase in a participant’s F1-score with respect to the number of iterations. Each iteration corresponded to an update from a single batch containing 256 training samples. (**Left** panel): LOSOCV trained using only data from the Stroke group. (**Right** panel): LOSOCV trained using ND + Stroke data. StrokeRH represents stroke patients with right hemiparesis, StrokeLH represents stroke patients with left hemiparesis.

**Figure 7 sensors-24-01618-f007:**
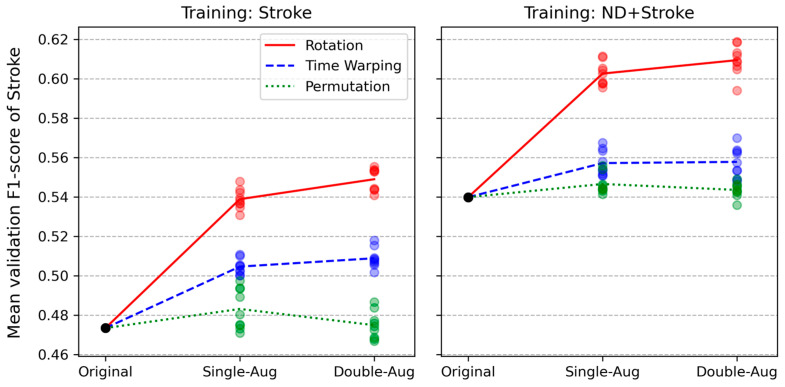
Effect of data augmentation on the Stroke (**left** panel) and ND + Stroke (**right** panel) baselines. Each colored, semi-transparent dot represents a mean F1-score of a LOSOCV run. The black dot (“original”) represents the baseline result for each baseline. The lines connect the baseline result to the mean of each augmentation result. Single-Aug represents single augmentation, Double-Aug represents double augmentation.

**Figure 8 sensors-24-01618-f008:**
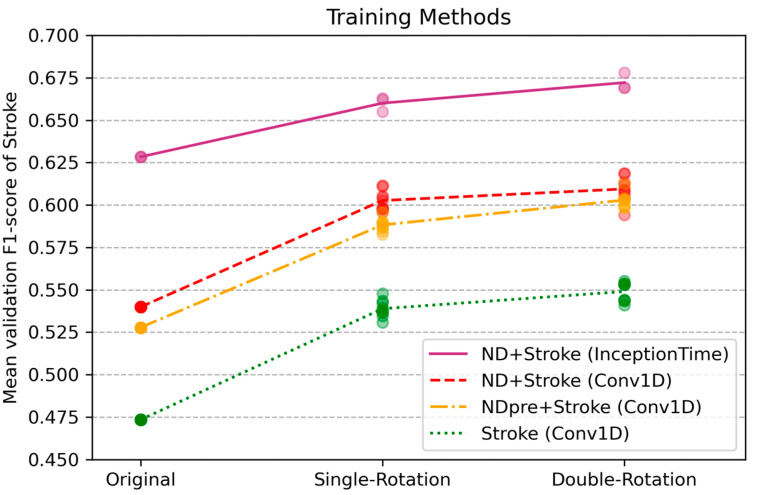
Comparison of some training methods. A coloured dot indicates the mean F1-score of each experiment run. Original: no augmentation applied, Single-Rotation: rotation applied to Stroke data and original data is augmented, Double-Rotation: two sets of differently rotated data augmented. ND + Stroke: both ND and Stroke data used as a baseline. NDpre + Stroke: Conv1D model first pre-trained with ND data, then transfer learning applied using Stroke data. Stroke: trained with Stroke data as a baseline.

**Figure 9 sensors-24-01618-f009:**
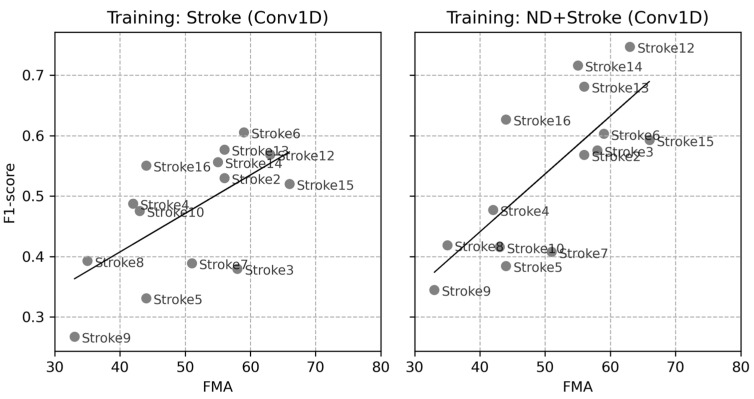
Correlation between FMA and individual F1-scores in the two baseline LOSOCVs. (**Left** panel): training with Stroke data only, (**Right** panel): training with ND + Stroke data.

**Figure 10 sensors-24-01618-f010:**
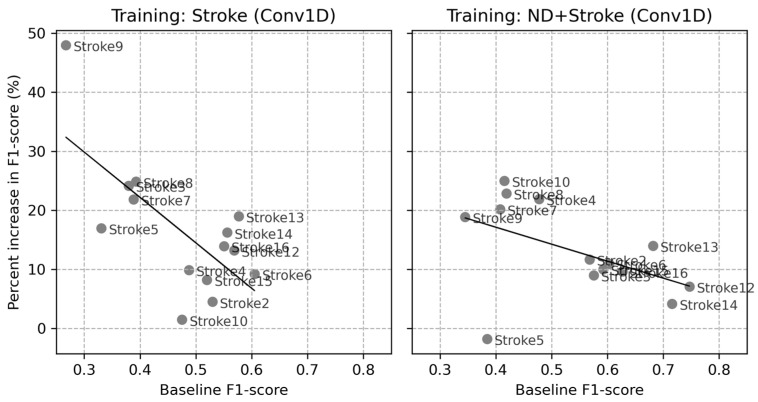
Correlation between the baseline F1-scores and the percentage increase in F1-scores in the two baseline LOSOCVs. (**Left** panel): training with Stroke data only, (**Right** panel): training with ND + Stroke data. The percentage increase was calculated based on the mean increase caused by both single and double rotational augmentations.

**Figure 11 sensors-24-01618-f011:**
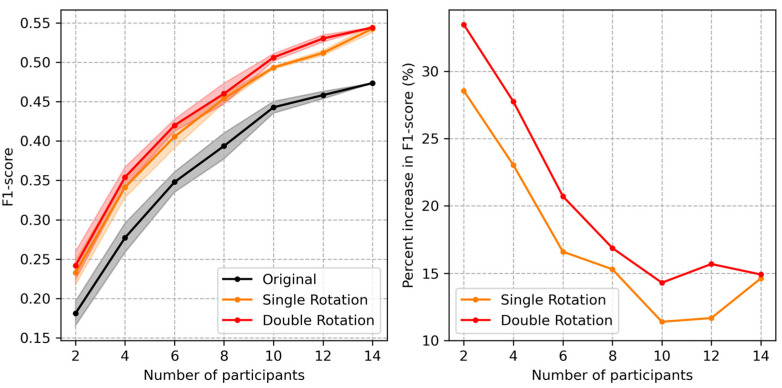
Change in F1-score with respect to varying numbers of training subjects in the Stroke group. Original: trained with subsets of the Stroke group. Single rotation: trained with rotationally augmented data (augmented only corresponding to the included subset). Double rotation: trained using two differently augmented datasets. (**Left** panel): The solid lines represent the mean and the shaded areas represent ± standard deviation ranges over multiple LOSOCV runs for each condition. (**Right** panel): Percentage increase of the augmented datasets relative to the original training sets.

**Figure 12 sensors-24-01618-f012:**
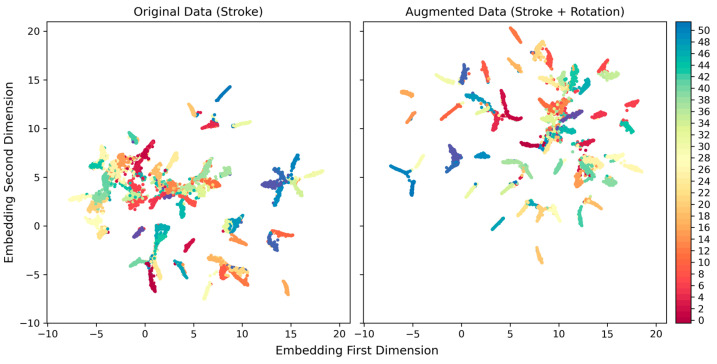
UMAP embedding of the final dense layer before the softmax layer in Conv1D model. (**Left** panel): Embedding of the model trained with the original Stroke data, (**Right** panel): embedding of the rotationally augmentated model. The colored dots represent the 52 classes.

**Table 1 sensors-24-01618-t001:** Mean F1-scores of the participants in each combination of training group and evaluation group.

Training Group	Evaluation Group			
	ND	Stroke	StrokeRH	StrokeLH
Trained using Stroke data	–	47.3 ± 10.4% *	47.2 ± 11.6%	47.6 ± 9.1%
Trained using ND + Stroke	90.5 ± 4.6%	54.0 ± 13.1%	56.9 ± 14.2%	48.7 ± 10.1%

* The number after “±” represents the standard deviation of individual F1-scores in the corresponding Stroke group or its subsets (StrokeRH and StrokeLH).

**Table 2 sensors-24-01618-t002:** Mean F1-scores of the participants in each combination of augmentation types and amount of augmented data.

Augmentation Type	Amount of Augmented Data					
	Stroke	Stroke + Single Augmentation	Stroke + Double Augmentation	ND + Stroke	ND + Stroke + Single Augmentation	ND + Stroke + Double Augmentation
Rotation	47.3%	53.9 ± 0.5% *	54.9 ± 0.5%	54.0%	60.3 ± 0.6%	60.9 ± 0.7%
Time-warping		50.5 ± 0.4%	51.0 ± 0.5%		55.7 ± 0.6%	55.8 ± 0.7%
Permutation		48.3 ± 1.0%	47.5 ± 0.7%		54.7 ± 0.5%	54.4 ± 0.4%

* The number after “±” represents standard deviation of mean F1-scores over nine LOSOCV runs.

**Table 3 sensors-24-01618-t003:** Mean F1-scores of the Train on Synthetic, Test on Real (TSTR) data evaluation. The original data were inserted as a reference.

Training Data			
Original (Stroke)	Rotation	Permutation	Time Warping
47.3%	49.8 ± 0.5% *	47.7 ± 0.7%	49.2 ± 0.8%

* The number after “±” represents standard deviation of mean F1-scores over nine LOSOCV runs.

**Table 4 sensors-24-01618-t004:** Mean F1-scores of the participants in each combination of training groups × evaluation groups.

Training Method	Amount of Augmented Data		
	Original	Single-Rotation	Double-Rotation
Stroke (Conv1D)	47.4%	53.9 ± 0.5% *	54.9 ± 0.5%
NDpre + Stroke (Conv1D)	52.8%	58.8 ± 0.4%	60.3 ± 0.4%
ND + Stroke (Conv1D)	54.0%	60.3 ± 0.6%	60.9 ± 0.7%
ND + Stroke (InceptionTime)	62.8%	66.0 ± 0.4%	67.2 ± 0.4%

* The number after “±” is standard deviation of mean F1-scores across multiple LOSOCV runs.

**Table 5 sensors-24-01618-t005:** Mean F1-scores of the OPPORTUNITY dataset with data augmentation.

Model	Data Augmentation			
	Original	Rotation	Permutation	Time Warping
DanHAR	60.5%	59.4 ± 0.7% *	61.7 ± 0.6%	65.0 ± 1.0%
InceptionTime	60.0%	56.2 ± 0.6%	59.4 ± 1.2%	62.9 ± 1.3%

* The number after “±” is standard deviation of mean F1-scores across multiple LOSOCV runs.

## Data Availability

The data presented in this study are available at https://github.com/youngminoh7/JU-IMU (accessed on 28 February 2024).

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
