# Peer review of "Data Augmentation Techniques for Accurate Action Classification in Stroke Patients with Hemiparesis"

_sensors, 2024, doi:10.3390/s24051618_

Round 1

Reviewer 1 Report

Comments and Suggestions for Authors

The investigation of how data augmentation may improve stroke classification results is an interesting issue, but there is a lack of originality in the presented study. 

Specific comments:

- It is well known in the literature that for physiological data the intrinsic component of the subject is very important. When a data augmentation technique is used, this should be done at the subject level, as well as the model learning step. In other words, mixing data segments from different signals allows the model to learn this intrinsic component, and it generally fails to perform well for new subjects. This issue has to be clarified in the revised version. 

- In order to be significant, the benchmark has also to involve wider comparisons with the state of the art methods for stroke detection. Conclusions may therefore be relevant beyond the used model.

- What about considering the problem as an anomaly detection issue? This would decrease the need of data augmentation since only the nominal class may be targeted. 

Minor comments:

- Please define acronym JU-IMU in the abstract

- Page 5: please better explain how rotation is done for these 1D signals. 

Author Response

Dear Reviewer,

Thank you for your time in reviewing my manuscript.

Please see the attachment for point-to-point responses.

Best regards,

Youngmin Oh

Reviewer 2 Report

Comments and Suggestions for Authors

The paper showed improved accuracy of action clarifications by stroke patients. However, the clinical relevance is not clear. The authors linked the F1-score with the Fugl-Meyer Assessment scores. "a higher F1-score from the model classification could be an indicator that a patient’s movement patterns are closer to the ND population, and vice versa." Do the authors try to use the F1-score as the index of patient recovery? If so, how? Please elaborate how the proposed method can improve the healthcare and recovery of stroke patients.

Minor concerns:

1. Line 32:  Is "use-dependent" or "user-dependent"?

2. Line 55: Please use the authors name as the subject of the sentence, not [10]

3. Line 124: Please clarify how the scales are determined.

4. Line 144: Please provide the reference of "previous study"

5. Line 322: Change" minute" to "small".

Author Response

(The authors gave the same response as above.)

Reviewer 3 Report

Comments and Suggestions for Authors

This research applied data augmentation and transfer learning on the JU-IMU dataset to verify whether data augmentation(rotation, permutation, and time warping) could improve model performance on classifying ADL performance through joint training signals.

The obvious limitation is the small sample size and lack of an external testing dataset.

Furthermore, there are some issues in the articles.

General:
1.  Some sentences with reference in the begining are lack of subjects. They should start with "The previous study"/"XXX et al.".

Introduction:
2. Line 38-41 Duplicate sentensce"Wide availability of wearable sensors such as smart watches or smart bands make them a good candidate for hardware of such a feedback system. The widespread availability of wearable sensors, such as smartwatches or smart bands, makes them suitable hardware candidates for such a feedback system"

Materials and Methods
Data Augmentation

3. The authors use rotation augmentation between -90 degrees to +90 degrees. However, in true world, this condition won't happen. Could the authors explain how they chose the angle?

Training and Evaluation

4. Line 217: How do the authors choose Participants 2, 4, 6, 8, 10, and 12 as the evaluation dataset?

5. Line 219: How do the authors choose Participants 2,3,6 to sample multiple times and mix with the ND group as an evaluation dataset?

Discussion

6. The strengths and limitations of the study should be stated.

7. If the authors could explain the biological meaning of data augmentation, the article could be more compact.

8. In line 484, what are the few studies?

Comments on the Quality of English Language

Many sentences are not complete. Some duplicated wordings were noted.

Author Response

(The authors gave the same response as above.)

Round 2

Reviewer 1 Report

Comments and Suggestions for Authors

The authors adequately addressed mot of my comments. There is only a remaining one regarding benchmark: my comment was about benchmark with other models, and not only with other databases.

Author Response

Dear reviewer,

Thank you once again for your valuable comments aimed at improving the manuscript. Regarding your comment about benchmarking with other models, I would like to provide the following response.

To begin, I have already added a new model, DanHAR, in the current revision, and it has been compared with InceptionTime in Table 5.

In this study, the focus has been on the systematic investigation of various data augmentation methods on a sparse stroke dataset. Given the limited availability of public datasets specifically designed for stroke upper extremity classification, a direct benchmark against other state-of-the-art (SOTA) models in the same domain was not feasible. Instead, comparisons were made with two prominent SOTA models from related but different domains: InceptionTime, a leading model for the UCR Time Series dataset, and DanHAR, known for its effectiveness in human action recognition tasks such as the OPPORTUNITY dataset.

The inclusion of InceptionTime and DanHAR in the evaluation serves to demonstrate the potential transferability and effectiveness of the proposed data augmentation methods across diverse tasks. InceptionTime's success in the UCR Time Series dataset and DanHAR's performance in human action recognition highlight the adaptability of these methods.

Looking forward, the aspiration is for the primary dataset used in this study, JU-IMU, to serve as a new benchmark for stroke action classification. With further research, it is anticipated that rigorous benchmarking on this dataset against a variety of models will become feasible, paving the way for more comprehensive evaluations and comparisons.

The inclusion of these SOTA models and the hope for JU-IMU to become a benchmark dataset signal the broader implications of this work within the field of stroke action classification. It is envisioned that these findings not only contribute to the understanding of data augmentation in sparse datasets but also provide a foundation for future research in similar domains.

I hope that this response adequately addresses your comment.

Thank you.

Best regards,

Youngmin Oh

Reviewer 2 Report

Comments and Suggestions for Authors

All concerns are addressed.

Author Response

Dear reviewer,

Thank you once again for your kind comments.

Your feedback has greatly assisted in improving the initial manuscript.

Best regards,

Youngmin Oh

Reviewer 3 Report

Comments and Suggestions for Authors

The authors did their best effort to refine the article and state the limitations of the current study.

Author Response

(The authors gave the same response as above.)
